# The use of complementary and alternative medicine among hypertensive and type 2 diabetic patients in Western Jamaica: A mixed methods study

Omolade Adeniyi[1‡], LaTimberly Washington[1‡], Christina J. Glenn[2], Sarah G. Franklin[1], Anniecia Scott[3], Maung Aung[3‡], Soumya J. Niranjan[4‡], Pauline E. Jolly[1‡*]

1 Department of Epidemiology, School of Public Health, University of Alabama at Birmingham, Birmingham, Alabama, United States of America, 2 Department of Biostatistics, School of Public Health, University of Alabama at Birmingham, Birmingham, Alabama, United States of America, 3 Epidemiology Unit, Western Regional Health Authority, Ministry of Health, Montego Bay, St. James, Jamaica, 4 Health Services Administration, School of Health Professions, University of Alabama at Birmingham, Birmingham, Alabama, United States of America

‡ OA and LW are Joint First authors, MA, SJN and PEJ are Joint Senior Authors to this work.
* jollyp@uab.edu

**Data Availability Statement:** All relevant data are within the paper and its Supporting Information files.

## Abstract

### Background

The simultaneous or intermittent use of alternative treatments and prescription medications for hypertension and type 2 diabetes mellitus can have adverse health effects.

### Objectives

To identify beliefs and practices associated with the use of alternative treatments for hypertension and type 2 diabetes mellitus among patients.

### Methods

A mixed-methods study including an investigator-administered survey and focus group discussion sessions using convenience sampling was conducted among patients aged ≥18 years during May to August 2018. Descriptive statistics were used to describe and compare demographic characteristics among groups of survey participants using JMP Pro 14.0. Thematic analysis was conducted to analyze the qualitative data using NVivo.

### Results

Most study participants (87–90%) were on prescription medication for their condition. Of survey participants, 69% reported taking their medication as prescribed and 70% felt that prescription medicine was controlling their condition. Almost all participants (98%) reported using alternative treatments, mainly herbal medications, and 73–80% felt that herbal medicines controlled their conditions. One-third believed that herbal medicines are the most effective form of treatment and should always be used instead of prescription medication.

**Funding:** This study was supported by the Minority Health International Research Training (MHIRT) grant no. T37-MD001448 from the National Institute on Minority Health and Health Disparities, National Institutes of Health (NIH), Bethesda, Maryland, USA, and the Western Regional Health Authority, Ministry of Health, Jamaica.

**Competing interests:** The authors have declared that no competing interests exist.

However, most participants (85%) did not believe that prescription and herbal treatments should be used simultaneously. Most (76–90%) did not discuss herbal treatments with their healthcare providers. Four themes emerged from the focus group sessions: 1) Simultaneous use of herbal and prescription medicine was perceived to be harmful, 2) Patients did not divulge their use of herbal medicine to healthcare providers, 3) Alternative medicines were perceived to be highly effective, and 4) Religiosity and family elders played key roles in herbal use.

## Conclusions

This study provides useful insights into perceptions and use of alternative treatments by patients that can be used by healthcare providers in developing appropriate interventions to encourage proper use of prescription medicines and alternative medicines resulting in improved management of these chronic diseases.

## Introduction

The World Health Organization (WHO) reports that Type 2 Diabetes Mellitus (T2DM) and Hypertension (HTN) are two of the top ten causes of death among the Jamaican population [1]. Worldwide, the prevalence of diabetes has increased significantly since 1995 with over 48 million people currently living with the disease. In Jamaica, it is projected that there will be an additional 33,000 people diagnosed with diabetes by the year 2030, with the current prevalence being at over 200,000 cases [2]. Jamaica's population is approximately 2.7 million people, and non-communicable diseases (NCDs) have increased the economic burden due to the working population being the most affected and decreasing productivity [3].

Complementary and alternative medicine (CAM) has been used for centuries across the globe and consist of a diverse subset of therapies such as dietary supplements, botanicals, traditional Chinese medicine, acupuncture, mind-body medicine, and therapeutic massage [1, 4, 5]. According to the World Health Organization, roughly 80% of the world's population use at least one form of CAM [6]. Reasons for CAM use vary by country and level of conventional healthcare available among the populations [6]. In countries where many individuals lack access to healthcare resources, and with increasing healthcare costs globally, CAM can often provide a more affordable and accessible alternative to conventional medical care [5]. In high income countries such as the United States of America, CAM use deviates from traditional practices and has been adopted from other countries where CAM consumption is within the dominant structure of healthcare. Types of CAM used in the Caribbean are often methods that have been practiced for generations and have deep cultural and/or religious roots [7]. A survey that examined the use of herbal remedies among rural and urban Jamaicans of varying socio-economic groups found that 100% of the participants used herbs [8].

Other research suggests that there are significant associations with herbal use and education, gender, religion, and health insurance status among Jamaicans [9]. It has also been shown that the use of herbal medicines in conjunction with prescription medications is common. Jamaicans are likely to use herbal medicine not only for the treatment of HTN and T2DM but also for illnesses such as the common cold, headache, or diarrhea [9]. In Jamaica and the United States, cases of HTN and T2DM are linked to lifestyle practices; however, it is evident that adherence to prescription medications for chronic diseases is lower for those living in the Caribbean compared to the United States [10].

This mixed-methods research study was conducted to provide a deeper understanding of CAM use for HTN and T2DM among Jamaicans. With the expected increase in T2DM and HTN, understanding the beliefs and use of alternative treatments is essential for the appropriate guidance of patients for proper management of these chronic diseases. We investigated CAM use, beliefs regarding the effectiveness of prescription medication and CAM, and discussion of CAM with healthcare providers (HCPs) by patients. The qualitative portion of the study provided an opportunity for open narratives and richer context within our target population.

## Material and methods

### Study design, site, and study population

A cross-sectional mixed-methods study was conducted from May to August 2018 in which convenience sampling was used to recruit patients ≥18 years of age attending clinics for HTN and T2DM in the four parishes of western Jamaica (St. James, Westmoreland, Hanover, and Trelawny) under the Western Regional Health Authority (WRHA). An exploratory design was used for the quantitative portion of the study among 60 participants [11]. Prior studies suggest the sample size (N = 60) is sufficient since the exploratory nature of the quantitative survey is the first stage of data collection and provides a rationale for defining future hypotheses for other stages of study [12]. In the concurrent quantitative-qualitative design, a smaller qualitative data design sample (N = 25) was determined as sufficient [11]. Clinic nurses informed patients of the study when they came in for an appointment. Patients who indicated interest to participate were introduced to the research team who told them about the study in private rooms in the clinic. After the potential participants were allowed to read the consent form, ask questions, and were satisfied that they wanted to participate, they were asked to sign the informed consent. At recruitment, participants were asked if they would be able to participate in a FGD session that would be arranged for a later date. Those who said that they could were asked to give their phone numbers to be contacted. Twenty-five participants completed the FGD sessions. Focus group participants were not allowed to complete the quantitative survey. Sixty participants completed the quantitative study.

### Inclusion and exclusion criteria

Participants in this study were adults (≥18 years of age) who had been diagnosed with HTN and/or T2DM and were attending health clinics located in one of the four parishes under the WRHA. Those who did not meet these criteria were excluded.

**Development of survey, focus group discussion guide and data collection.** A survey was developed to collect data on demographic factors (age, education, employment status, income, and residence), use and beliefs regarding the effectiveness of prescription medication for HTN and T2DM, use and beliefs regarding the effectiveness of CAM, concomitant use of prescription medicine and CAM, and discussion of CAM with HCPs. Questions on knowledge, use and beliefs of CAM, and concomitant use of CAM with prescription medicines were adapted from questions used in published papers on studies conducted on CAM in Jamaica and Trinidad and Tobago [6, 8, 9, 13]. Questions were added to allow participants to list the types of CAM and the main herbal medicines they were using for HTN or T2DM. The survey was reviewed by Jamaican HCPs and revised. It was then pilot tested among 10 clinic patients in Jamaica similar to the ones recruited for the study and again revised before use. Other validation methods were not conducted. Surveys took approximately 30–45 minutes to complete.

Questions on the FGD guide were adapted from pertinent questions on the survey to generate discussion on the use of CAM, perception of the effectiveness of CAM and prescription

medication, concurrent use of prescription medication and CAM, and communication with HCPs regarding medicinal practices. Five FGD sessions (two in St. James and one in each of the other three parishes) were conducted with 25 participants (three with 5, one with 4 and one with 6 participants). Participants who indicated that they could participate in a FGD session and gave their telephone numbers at recruitment were contacted with the date and time that the FGD session would be held. FGD sessions were conducted in vacant conference and exam rooms at the clinics. Each FGD session lasted approximately 75 minutes and was comprised of male and female participants who had HTN, T2DM, or both. Demographic information was obtained for each participant and FGD questions were tailored to generate discussion on when participants chose to use CAM, their perception of the effectiveness of CAM versus prescription medication, whether participants used prescription medication and CAM concurrently, and communication with their HCPs regarding their medicinal practices.

## Ethical approval

The study was approved by the Institutional Review Board at the University of Alabama at Birmingham and by the Western Regional Health Authority; protocol approval #IRB-170310006.

## Data analysis

**Quantitative data.** The quantitative data for the 60 survey respondents were entered into excel and imported into JMP Pro 14.0 for analysis. Descriptive statistics were used to describe demographic characteristics of participants using mean ± standard deviation for continuous variables and frequency (percentage) for categorical variables. Demographic characteristics between disease groups were compared using a Fisher's Exact test for categorical characteristics, and an analysis of variance for continuous characteristics. The significance level for these comparisons was set at $p \leq 0.05$.

**Qualitative data.** Transcripts from the five FGD sessions were reviewed by three independent coders (SJN, LW, and OA) and coded using QSR International's NVivo 11.4.3 software using line-by-line coding of all responses to the FGD questions, followed by focused coding for directed codes. We utilized constant comparative method to generate themes from the transcribed data [11]. Trustworthiness was achieved through data triangulation and peer debriefing [14]. Themes are presented in a manner that convey understanding of CAM and prescription medication use.

## Results

### Quantitative results

**Demographic characteristics of survey participants.** Sixty participants, aged 35–82 years, completed the survey; of those 60, 37 (61.7%) had HTN only, 10 (16.7%) had T2DM only, and 13 (21.7%) had both diseases (Table 1). On average, patients were 59.5 years; most were female (75%), had a secondary education (55%), had no income (28.8%), or were earning <J\$24,800 a month (30.5%) (USD 1 = JD 122 at the time of the study; Table 1). Marital status was significantly associated with disease group, p = 0.0234; all other demographic characteristics did not significantly differ by disease group, all p>0.05, Table 1.

**Practices and beliefs regarding prescription medication use for T2DM and HTN (Table 2).** There were no significant associations among disease groups and survey responses, all p>0.05 (Table 2). More than half of participants (58.3%) had been diagnosed with T2DM or HTN for over 10 years and all but two (1 with HTN; 1 with HTN+T2DM) reported using CAM (Table 2). The most common CAM method used was herbal medicine;

**Table 1. Demographic characteristics of survey participants by disease group.**

| | Total (N = 60) | T2DM (N = 10) | HTN (N = 37) | HTN+T2DM (N = 13) | p-value |
|---|---|---|---|---|---|
| **Characteristic** | | | | | |
| **Age, Mean (SD) years** | 59.5 (11.2) | 62.3 (11.0) | 59 (11.4) | 58.8 (11.2) | 0.6920[d] |
| **Sex, N(%)** | | | | | 0.4874 |
| Male | 15 (25.0) | 4.0 (40.0) | 8.0 (21.6) | 3.0 (23.1) | |
| Female | 45 (75.0) | 6.0 (60.0) | 29.0 (78.4) | 10.0 (76.9) | |
| **Marital Status, N(%)** | | | | | 0.0234 |
| Married[a] | 29 (41.7) | 8.0 (80.0) | 10.0 (27.0) | 7.0 (53.8) | |
| Single | 25 (10.0) | 2.0 (20.0) | 21.0 (56.8) | 6.0 (46.2) | |
| Other[b] | 6 (48.3) | 0.0 (0.0) | 6.0 (16.2) | 0.0 (0.0) | |
| **Education, N(%)** | | | | | 0.3911 |
| None | 4 (6.7) | 1.0 (10.0) | 3.0 (8.1) | 0.0 (0.0) | |
| Some/Complete Primary | 20 (33.3) | 3.0 (30.0) | 11.0 (29.7) | 6.0 (46.2) | |
| Some/Complete Secondary | 33 (55.0) | 6.0 (60.0) | 22.0 (59.5) | 5.0 (38.5) | |
| College/University | 3 (5.0) | 0.0 (0.0) | 1.0 (2.7) | 2.0 (15.4) | |
| **Parish, N(%)** | | | | | 0.6677 |
| Hanover | 15 (25.0) | 2.0 (20.0) | 9.0 (24.3) | 4.0 (30.8) | |
| St. James | 12 (20.0) | 4.0 (40.0) | 5.0 (13.5) | 3.0 (23.1) | |
| Trelawny | 14 (23.3) | 2.0 (20.0) | 9.0 (24.3) | 3.0 (23.1) | |
| Westmoreland | 19 (31.7) | 2.0 (20.0) | 14.0 (37.8) | 3.0 (23.1) | |
| **Income[c], N(%)** | | | | | 0.6466 |
| None | 17 (28.8) | 3.0 (30.0) | 8.0 (22.2) | 6.0 (46.2) | |
| <J$24,800 | 18 (30.5) | 3.0 (30.0) | 13.0 (36.1) | 2.0 (15.4) | |
| J$24,801-J$60,000 | 16 (27.1) | 2.0 (20.0) | 11.0 (30.6) | 3.0 (23.1) | |
| >J$60,000 | 8 (13.6) | 2.0 (20.0) | 4.0 (11.1) | 2.0 (15.4) | |

HTN = Hypertension, T2DM = Type 2 Diabetes Mellitus.

[a] Includes Common Law marriage

[b] Other includes divorced, widowed and separated

[c] One participant declined to comment

[d] Analysis of Variance. Fisher's Exact p-values for all categorical variables; p-value ≤0.05 considered meaningful.

this was followed in sequential order by exercise, spiritual healing, relaxation techniques, and diet modifications.

Most participants (90% with T2DM, 86.5% with HTN, and 100% with HTN+T2DM) were on prescription medication for their condition and most received their medication from a pharmacy at a health center or a private pharmacy (Table 2). Most participants (73.3%) reported filling their prescriptions on time. The main difficulties participants reported that they experienced in picking up their prescriptions were financial difficulty, issues with their government-issued insurance card, the pharmacy being out of medication, transportation issues, and need for someone to pick up the medication. T2DM (44.4%), HTN (71.9%) participants and those with both conditions (92.3%) reported getting their medication free of cost.

Participants with T2DM (60%), HTN (68.8%), and both conditions (69.2%) reported taking their medication as prescribed. Side effects, substitution/preference of alternate medication, forgetting to take, and stopping to see if the medication was working were the main reasons given by participants for not taking their medication as prescribed. Seventy-three percent of participants felt that their prescription medication was controlling their condition; most of

**Table 2. Practices regarding prescription medication usage by survey participants.**

| | T2DM (N = 10) | HTN (N = 37) | HTN+T2DM (N = 13) | p-value |
|---|---|---|---|---|
| Survey Question, N (%) | | | | |
| **When were you diagnosed with T2DM/HTN?** | | | | 0.1780 |
| ≤10 years ago | 3.0 (30.0) | 19.0 (51.4) | 3.0 (23.1) | |
| >10 years ago | 7.0 (70.0) | 18.0 (48.6) | 10.0 (76.9) | |
| **Are you currently using any alternative treatments/home remedies for T2DM/HTN?** | | | | 0.6237 |
| Yes | 10.0 (100.0) | 36.0 (97.3) | 12.0 (92.3) | |
| No | 0.0 (0.0) | 1.0 (2.7) | 1.0 (7.7) | |
| **Are you currently on medication from the clinic or doctor for your T2DM/HTN?** | | | | 0.4006 |
| Yes | 9.0 (90.0) | 32.0 (86.5) | 13.0 (100.0) | |
| No | 1.0 (10.0) | 5.0 (13.5) | 0.0 (0.0) | |
| **Do you have difficulty picking up your T2DM/HTN medicine?** | | | | 0.4768 |
| Yes | 1.0 (10.0) | 11.0 (33.3) | 3.0 (23.1) | |
| No | 8.0 (80.0) | 20.0 (60.6) | 10.0 (76.9) | |
| Sometimes | 1.0 (10.0) | 2.0 (6.1) | 0.0 (0.0) | |
| **Do you get your T2DM/HTN medication free of cost?** | | | | 0.0503 |
| Yes | 4.0 (44.4) | 23.0 (71.9) | 12.0 (92.3) | |
| No | 5.0 (55.6) | 9.0 (28.1) | 1.0 (7.7) | |
| **How much do you pay for your T2DM/HTN medicine?** | | | | 0.1633 |
| <J$2000 | 1.0 (12.5) | 3.0 (17.7) | 3.0 (37.5) | |
| J$2000-J$4999 | 6.0 (75.0) | 8.0 (47.1) | 1.0 (12.5) | |
| ≥J$5000 | 1.0 (12.5) | 6.0 (35.3) | 4.0 (50.0) | |
| **Do you refill your T2DM/HTN medication on time?** | | | | 0.4366 |
| Yes | 9.0 (90.0) | 23.0 (71.9) | 12.0 (92.3) | |
| No | 0.0 (0.0) | 6.0 (18.8) | 1.0 (7.7) | |
| Sometimes | 1.0 (10.0) | 3.0 (9.4) | 0.0 (0.0) | |
| **Do you always take your T2DM/HTN medicine(s) as prescribed?** | | | | 0.1795 |
| Yes | 6.0 (60.0) | 22.0 (68.8) | 9.0 (69.2) | |
| No | 2.0 (20.0) | 10.0 (31.2) | 3.0 (23.1) | |
| Sometimes | 2.0 (20.0) | 0.0 (0.0) | 1.0 (7.7) | |
| **Do you experience any side effects from your T2DM/HTN medicine(s)?** | | | | 0.9999 |
| Yes | 4.0 (40.0) | 15.0 (46.9) | 6.0 (46.2) | |
| No | 6.0 (60.0) | 17.0 (53.1) | 7.0 (53.8) | |
| **If you get a normal blood pressure reading, do you stop taking your prescription medication?** | | | | 0.9999 |
| Yes | NA | 6 (18.7) | 10 (77.0) | |
| No | NA | 26 (81.3) | 3 (23.0) | |
| **Does a normal BP (120/80) mean that:** | | | | 0.7037 |
| You are cured | NA | 1 (2.7) | 0 (0.0) | |
| Your blood pressure is normal at the time but you still have high blood pressure. | NA | 36 (97.3) | 13 (100.0) | |

Fisher's Exact p-values; p ≤ 0.05 considered meaningful. HTN = Hypertension; T2DM = Type 2 Diabetes Mellitus.

those who felt the medication was not controlling their condition said that the medication was not effective enough. The most common side effects reported by participants with T2DM were itchiness, stomach pains, and increased urination and by participants with HTN were dizziness, headache, nausea/stomachache, muscle pain, and increased urination. The majority of participants with HTN (97.3%) and both HTN and T2DM (100%) knew that a blood pressure reading of 120/80 did not mean that they were cured or that they should stop taking their prescription medication.

Table 3 shows the total number of participants who selected each herb as well as the total numbers stratified by disease group. The top five herbs listed by participants with T2DM were guinea hen (*Petiveria alliacea*), moringa (*Moringa oleifera*), garlic (*Allium sativum*), ginger (*Zingiber officinale*), and turmeric (*Curcuma longa*). Participants with HTN listed garlic, moringa, guinea hen, lime (*Citrus aurantiifolia*), and ginger as their top five and those with both T2DM and HTN listed turmeric, moringa, ginger, lime, and garlic.

**Knowledge, attitudes, and practices regarding CAM use by disease group (Table 4).**
Participants were asked to respond to a variety of questions related to their perceptions and behaviors regarding CAM use. Participant responses were not significantly associated with disease group, all p>0.05 (Table 4). Participants (73%) indicated that herbal medicines controlled their conditions. T2DM (30%), HTN (42.4%), and participants with both conditions (50%) reported that they do not take their prescription medication as prescribed when they take herbal medicines. The reasons given for not taking herbal and prescription medicines simultaneously were: did not want herbal medicine to interfere with prescription medicine, did not want to take too much medicine, preferred to use herbs, wanted to take less prescription medicine, wanted to see if herbs were more effective, and did not want blood pressure to drop too low. All of the diabetic participants and those with both conditions along with 84% of hypertensive participants reported that they had received information about CAM, mainly that CAM can help to control T2DM and HTN, can benefit and are good for the body, and can kill cancer cells. A few participants reported that they heard that herbs were not good for the body and can worsen symptoms. Most reported that they received information from family, friends, and community members; only small percentages (10% with T2DM, 24% with HTN, 46.2% with both conditions) reported discussing CAM with their HCPs. The main reasons participants gave for not discussing CAM with HCPs were: HCPs do not ask about CAM, they did not think of discussing CAM with HCPs, and HCPs do not approve of CAM. The top five reasons given in choosing to use CAM were: CAM helps to control blood sugar/blood pressure, others recommended CAM, CAM is used when they do not have their prescription medication, they wanted to try CAM, and CAM is preferred over prescription medication. About one-quarter of participants reported taking CAM once or twice daily and when they are experiencing symptoms or are unable to afford prescription medication. Over 90% of participants reported no negative side effects of CAM, a few reported sleepiness, dizziness, loss of balance, and sinus issues. Twenty percent of participants with T2DM, 27% with HTN, and 23.1% of those with both conditions felt that there were possible harmful effects of taking prescription medications and CAM simultaneously. Apart from general side effects such as stomachache

**Table 3. Total number of participants who selected each herb and total numbers stratified by disease group.** Herbs listed by highest percentage of total.

| Top six herbs listed by participants | Total (N = 60) | Type 2 Diabetes Mellitus (N = 10) | Hypertension (N = 37) | T2 Diabetes Mellitus and Hypertension (N = 13) |
|---|---|---|---|---|
| | N (%) | N (%) | N (%) | N (%) |
| Guinea Hen | 20 (33.3) | 8 (80.0) | 10 (27.0) | 2 (15.4) |
| Moringa | 20 (33.3) | 5 (50.0) | 11 (29.7) | 4 (30.8) |
| Garlic | 32 (53.3) | 3 (30.0) | 26 (70.3) | 3 (23.1) |
| Ginger | 13 (21.7) | 3 (30.0) | 6 (16.2) | 4 (30.1) |
| Turmeric | 12 (20.0) | 3 (30.0) | 4 (10.8) | 5 (38.5) |
| Lime | 11 (18.3) | 0 (0.0) | 7 (18.9) | 4 (30.8) |

Other herbs listed include were lemon grass/fever grass (*Cymbopogon citratus*), cerasse (*Momordica charantia*), mint (*mentha*), cinnamon (*Cinnamomum verum*), soursop leaves (*Annona muricata*), coconut water (*Cocos nucifera*), ganja (*Cannabis*), breadfruit leaves (*Artocarpus altilis*), rosemary (*Salvia Rosmarinus*), vervain (*Verbena officinalis*) and neem (*Azadirachta indica*).

Table 4. Knowledge, attitudes, and practices regarding CAM use by disease group.

| | T2DM (N = 10) | HTN (N = 37) | HTN+T2DM (N = 13) | p-value |
|---|---|---|---|---|
| **Survey Question, N (%)** | | | | |
| **Do you think herbal medicines control your T2DM/HTN?** | | | | 0.8679 |
| Yes | 7.0 (70.0) | 22.0 (68.8) | 11.0 (84.6) | |
| No | 2.0 (20.0) | 3.0 (9.4) | 1.0 (7.7) | |
| Sometimes | 1.0 (10.0) | 6.0 (18.8) | 1.0 (7.7) | |
| Unsure | NA | 1.0 (3.1) | 0.0 (0.0) | |
| **When you use herbal medicine, do you still take your prescription medication as prescribed?** | | | | 0.1174 |
| Yes | 7.0 (70.0) | 19.0 (57.6) | 4.0 (33.3) | |
| No | 3.0 (30.0) | 14.0 (42.4) | 6.0 (50.0) | |
| Sometimes | 0.0 (0.0) | 0.0 (0.0) | 2.0 (16.7) | |
| **Have you received information about alternative treatments?** | | | | 0.2074 |
| Yes | 10.0 (100.0) | 31.0 (83.8) | 13.0 (100.0) | |
| No | 0.0 (0.0) | 6.0 (16.2) | 0.0 (0.0) | |
| **Where or from whom did you receive the information?** | | | | 0.8447 |
| Word of Mouth | 6 (60.0) | 20 (66.6) | 11 (84.6) | |
| Internet, Television, or Radio | 2 (20.0) | 6 (20.0) | 2 (3.8) | |
| Own Research | 1 (10.0) | 2 (6.7) | 0 (0.0) | |
| Health Provider | 1 (10.0) | 2 (6.7) | 0 (0.0) | |
| **Have you discussed alternative treatments for T2DM/HTN with your healthcare provider?** | | | | 0.1553 |
| Yes | 1.0 (10.0) | 9.0 (24.3) | 6.0 (46.2) | |
| No | 9.0 (90.0) | 28.0 (75.7) | 7.0 (53.8) | |
| **Do you take alternate treatments when you cannot afford your prescribed medications?** | | | | 0.7336 |
| Yes | 2.0 (20.0) | 9.0 (25.7) | 4.0 (30.8) | |
| No | 7.0 (70.0) | 25.0 (71.4) | 8.0 (61.5) | |
| Sometimes | 1.0 (10.0) | 1.0 (2.9) | 1.0 (7.7) | |
| **Do you experience any negative side effects when taking alternative medication?** | | | | 0.7470 |
| Yes | 0.0 (0.0) | 3.0 (8.1) | 0.0 (0.0) | |
| No | 10.0 (100.0) | 34.0 (91.9) | 13.0 (100.0) | |
| **Are there any possible harmful effects of using both herbal and prescription medicines at the same time?** | | | | 0.9174 |
| Yes | 2.0 (20.0) | 9.0 (27.3) | 3.0 (23.1) | |
| No | 8.0 (80.0) | 20.0 (60.6) | 9.0 (69.2) | |
| Unsure | 0.0 (0.0) | 4.0 (12.1) | 1.0 (7.7) | |

Fisher's Exact p-values; p ≤ 0.05 considered meaningful. HTN = Hypertension; T2DM = Type 2 Diabetes Mellitus.

and headache, a few participants felt that taking prescription medication and CAM simultaneously could lower blood sugar or blood pressure too much.

**Participants' beliefs regarding the use of prescription medications and CAM (Table 5).** The responses from a series of questions that evaluated participants' beliefs regarding the use of prescription medication and CAM and using them simultaneously are presented in Table 5. No significant differences in beliefs about CAM by disease group were found, all p>0.05. Thirty-seven percent of participants believed that CAM should always be used instead of prescription medication and that CAM is the most effective form of treatment for their conditions. Seventy-five percent indicated that they would communicate negative side effects to their HCP before deciding to discontinue their prescription medications. A majority of participants (81.6%) did not believe that it was okay to use prescription and CAM at the same time and 80% believed that they should always discuss CAM use with their HCP.

**Table 5. Perceptions and beliefs regarding use of alternative and prescription treatments by disease group.**

|  | T2DM (N = 10) | HTN (N = 37) | HTN+T2DM (N = 13) | p-value |
|---|---|---|---|---|
| **Survey Question, N (%)** |  |  |  |  |
| **Alternative treatments should always be used instead of prescription medication.** |  |  |  | 0.8835 |
| Yes | 4.0 (40.0) | 12.0 (32.4) | 6.0 (46.2) |  |
| No | 4.0 (40.0) | 16.0 (43.2) | 6.0 (46.2) |  |
| Sometimes | 1.0 (10.0) | 7.0 (18.9) | 1.0 (7.7) |  |
| Unsure | 1.0 (10.0) | 2.0 (5.4) | 0.0 (0.0) |  |
| **Alternative treatments are more effective at treating T2DM/HTN than prescription medication.** |  |  |  | 0.4118 |
| Yes | 5.0 (50.0) | 14.0 (37.8) | 3.0 (23.1) |  |
| No | 3.0 (30.0) | 19.0 (51.4) | 7.0 (53.8) |  |
| Sometimes | 1.0 (10.0) | 3.0 (8.1) | 3.0 (23.1) |  |
| Unsure | 1.0 (10.0) | 1.0 (2.7) | 0.0 (0.0) |  |
| **If you are experiencing unpleasant side effects of prescription medication, is it okay to stop taking the medicine without consulting your healthcare provider?** |  |  |  | 0.5566 |
| Yes | 1.0 (10.0) | 9.0 (24.3) | 4.0 (30.8) |  |
| No | 9.0 (90.0) | 28.0 (75.7) | 8.0 (61.5) |  |
| **Do you think it is okay to use both prescription medication and alternative treatments at the same time to treat T2DM/HTN?** |  |  |  | 0.1555 |
| Yes | 3.0 (30.0) | 4.0 (10.8) | 4.0 (30.8) |  |
| No | 7.0 (70.0) | 33.0 (89.2) | 9.0 (69.2) |  |
| **Do you think that you should always discuss any alternative treatments for your condition with your healthcare provider?** |  |  |  | 0.3763 |
| Yes | 7.0 (70.0) | 29.0 (78.4) | 12.0 (92.3) |  |
| No | 3.0 (30.0) | 8.0 (21.6) | 1.0 (7.7) |  |

Fisher's Exact p-values; p ≤ 0.05 considered meaningful. HTN = Hypertension; T2DM = Type 2 Diabetes Mellitus.

## Qualitative results

Four common themes emerged during FGD sessions (Table 6), which revealed the views and beliefs of participants regarding CAM and prescription medication use for chronic disease management.

**Theme #1: Simultaneous use of herbal and prescription medicine was perceived to be harmful.** With regard to the simultaneous use of prescription medications with herbal medicines, most participants indicated that simultaneous use would cause adverse health effects and that both should not be taken concomitantly. Some participants expressed the consequences of misusing these treatments, which include low blood pressure or low blood glucose and the possibility of losing consciousness. One participant expressed her personal and interpersonal experiences that summarized this theme:

> *I went to the clinic and someone told me to take it [garlic] like pill but don't use it with the medication because it will knock you out because it happened to me once [meaning to the other person] -(Participant 2, Trelawny, Female)*

**Table 6. Focus group themes.**

| |
|---|
| 1) Simultaneous use of herbal and prescription medicine was perceived to be harmful. |
| 2) Patients did not divulge their use of herbal medicine to healthcare providers. |
| 3) Alternative medicines were perceived to be highly effective |
| 4) Religiosity and family elders played key roles in herbal use. |

*When I used them at the same time it hit me out and sent me to the hospital and made the pressure very low, so I don't use them together.* -(Participant 2, Trelawny, Female)

Relating a similar experience, one participant discussed her regimen for taking prescription medication and herbs.

*If I could get some herbs to take and see it helping the sugar and the pressure, I will leave the medication, but you cannot just come off it quick because it will physically hurt you.* -(Participant 3, Trelawny, Female)

**Theme #2: Patients did not divulge their use of herbal medicine to healthcare providers.** Most participants conveyed that they did not feel it was necessary to communicate CAM use with their HCP. There was a perception of fear that the HCP would disagree with the patients' decision to use CAM and advise against what was working for the patients. There were also indications that because herb use is an intricate part of the Caribbean culture, some herbs were used for other conditions, such as headaches or sinuses and without the intention of treating an individual's HTN or T2DM. Therefore, participants did not think to inform their HCP. One participant stated,

*I'm not discussing any herbs with the doctor because when you tell him about herbs he says 'nonsense', 'foolishness' so I continue to drink the herb. I put the medication aside for a while, go on the herb, and then back to the medication.*—(Participant 3, Trelawny, Female)

On the other hand, the depth of the HCP-patient relationship was also an indicator of whether a respondent felt comfortable with disclosing information about herb use. There were several physicians working at the clinics that were not originally from Jamaica, which influenced the communication a patient received about herb use.

*My doctor is a Nigerian that supports both herbs and medication. I would love my Nigerian doctor to give me the herbs because I know he has an herb book and he knows which ones are best so I can get a list.*—(Participant 4, Trelawny, Female)

**Theme #3: Alternative medicines were perceived to be highly effective.** Most of our participants perceived CAM to be more effective than prescription medications due to the known possible side effects of prescription medication. They also believed that some prescription medication alleviated the chronic disease but caused other complications.

*Yes, I believe the herb is coming from the spiritual background the medication the doctor gives is not as effective. I discovered that even the medication slows down your sex organ and the herb uplifts it.*—(Participant 23, St. James, Male)

*I think so because a lot of the chemical treatment ends up damaging the body so I think more research should be done in the herbal area, a lot of doctors don't support it because they won't be getting money.*—(Participant 5, St. James, Male)

One participant conveyed that although she was in favor of CAM, her prescription medication was equally as effective and provided her with rapid results. When the group was asked if they still take their prescription medication as prescribed when using herbal medications, one participant answered:

*I live on my medication but I just take it (herb) for tea so I still take my [prescription] medication.*—(Participant 16, Westmoreland, Female)

**Theme #4: Religiosity and family elders played key roles in herbal use.** When participants were asked about how they were informed of CAM, most responded that their family and community elders influenced them and that they grew up witnessing people in their communities sharing CAM methods to treat a wide variety of illnesses. Also, many expressed listening to radio doctors explain CAM methods for certain conditions.

*I learned a lot from my grandmother even when I had my children, she told me what type of herbs to give them.*—(Participant 11, Hanover, Female)

*I heard from my parents my mother and father they said that the garlic is good for pressure.*–(Participant 14, Hanover, Male)

*I hear plenty people talk about the garlic plus a doctor I hear over the radio talks about garlic and says it is good.*—(Participant 1, Trelawny, Female)

Within each FGD session, participants expressed that religion played a role in their comfort with using CAM for chronic diseases and other illnesses. For instance, two participants responded:

*God put herbs on the earth to heal people, but they are not using it because when I take the pill they give me a lot of side effects.*—(Participant 3, Trelawny, Female)

*Herbal comes from my religion the Seventh-day Adventist, the prophetess, Ellen, tells us that herbs are good for the body*—(Participant 23, Hanover, Male)

## Discussion

Results from the quantitative and qualitative analysis mirror each other. Responses from our focus groups provided a better understanding of the societal beliefs that individuals have acquired and shed light on why some individuals prefer CAM methods, particularly herbal remedies, to prescription medication.

The survey revealed that a majority of our participants chose to use CAM because prescription medications had side effects that included headaches, decreased energy, and stomachaches. This theme was noted in our FGD sessions in which participants stated that the reduced or nonexistent side effects of CAM were preferred over side effects from prescription medicines. Participants also believed that since prescription medications were made from herbs they were the purest form of medication and best for the body. Studies conducted to understand the perceived risks and benefits of herbal use by participants, report that herbal use is not perceived as being safer than conventional medicines but are viewed as being more "natural" [15].

The majority of participants stated that they do not use herbal and prescription medicines together fearing that combined use will decrease their blood pressure or blood glucose to dangerous levels. Several randomized control trials have been conducted with different herbs and orthodox medications among both diabetic and hypertensive patients. Several trials have shown that garlic significantly lowered both systolic and diastolic blood pressure among patients who were administered garlic (extract or powder at concentrations of 240–960 mg/

day) plus antihypertensive drugs or diuretics compared to those administered placebo plus antihypertensive drugs or diuretics [16, 17]. No serious adverse events were reported from the use of garlic and prescription medicines in these studies. Garlic was also found to significantly reduce the level of fasting blood glucose in T2DM patients in several randomized control trials of patients treated with garlic and anti-diabetic drugs compared to control groups [18–21]. In these studies, significant improvements in total blood cholesterol, high density lipoprotein and low-density lipoprotein were also obtained after garlic administration. Heartburn was reported by some patients but was not significantly different between the treated and control groups.

In a clinical trial with ginger, one gram of ginger was shown to work synergistically with Nifedipine (10 mg daily) in inhibiting platelet aggregation in hypertensive patients. This shows that addition of ginger to hypertensive medicine was beneficial for cardiovascular and cerebro-vascular complication due to platelet aggregation [22]. No adverse effect of ginger was reported. However, studies of other herbs indicate that concomitant use with prescription medications may cause adverse reactions. Several studies reviewed by Izzo et al. (2005) showed adverse reactions between cardiovascular drugs such as warfarin and a number of herbs including garlic and ginkgo resulting in increased anticoagulation and with a number of other herbs such as ginseng and green tea causing decreased anticoagulation [23–27].

Consistent with other studies, our study participants also believed it is important to communicate their use of CAM with their HCPs, although they chose not to disclose CAM use due to fear of disapproval [8]. A study reported that HCPs do not have adequate knowledge of CAM use, and thus are not able to give medical advice to patients who use CAM [28]. This may in part explain HCPs' reluctance in discussing CAM use with patients. Moreover, literature also suggests that HCPs want to understand the use of CAM and the case-based research behind its use [29]. Therefore, changes to the curriculum for medical, nursing, dietetic and pharmacy students, residents, and fellows to include CAM methods may facilitate increased knowledge of the beneficial and harmful effects of CAM. A finding of interest was that nationality and cultural background of HCPs may influence how information on CAM methods is translated to the public. A participant discussed her interactions with a HCP of African nationality who she is convinced has good knowledge of herbs. Studies conducted in Trinidad and Tobago and Jamaica found a higher level of acceptance of CAM use among Trinidadian HCPs in comparison to those from Jamaica; however, HCPs in both countries seemed to lack proper knowledge about herb-drug use interactions that could be contributing to the lack of communication with patients [8].

Previous research suggests that there should be increased efforts aimed to increase information on the possible harmful effects of concurrent herb use with prescription drugs [6].

Religion was an important factor in influencing herbal use in this study. Therefore, clergy members could be used in conjunction with HCPs to relay pertinent information regarding CAM use. Recent studies show that clergy members are key components in health promotion [30].

## Limitations

This study has limitations that should be considered in interpreting the results. First, the convenience sampling method used is prone to inherent bias in representation. Some patients declined to participate in the study due to lack of trust in providing personal health information to the research staff; they feared that the personal information provided could be used against them. Secondly, the study sample represents only patients who attended the chronic disease clinics in the WRHA that were included in the study, therefore, the results may not be generalizable to patients attending other clinics under the WRHA or in other regions of

Jamaica that were not sampled. Additionally, the data were self-reported and as a result might be subject to social desirability bias.

## Conclusions

This study shows that CAM methods such as use of herbs, prayer, diet, and exercise to treat chronic diseases are a part of daily life for many Jamaicans. The findings from the survey and the themes from the FGD sessions provided new information and useful insights into the perceptions of participants regarding their inclination to use CAM. Participants believed based on their own personal experiences, and those of community/church members, that alternative treatments are effective with far less adverse effects as compared to prescription medicine. They did not discuss CAM use with their HCPs since they felt that most HCPs did not endorse the use of CAM. These findings highlight the importance of HCP-initiated conversations about patient use of CAM. Many participants are aware that there could be adverse reactions to the concurrent use of prescription and herbal treatments and developed their own algorithms of use which could be harmful. Thorough explanations of the effects of simultaneous or intermittent use of prescription and herbal treatments from knowledgeable HCPs is essential. Thus, the findings from this study indicate the need to include salient information on CAM in the professional curricula of HCPs and can be used in developing appropriate interventions to ensure the proper use of prescription medicines and CAM. This should result in improved management of T2DM and HTN among patients.

## Supporting information

**S1 Dataset. CAM dataset HTN & T2DM Jamaica.**
(XLSX)

**S1 File. Survey used in study.**
(PDF)

**S2 File. Focus group discussion guide.**
(DOCX)

## Acknowledgments

We thank the nurses in the clinics who facilitated the study and the patients who participated.

## Author Contributions

**Conceptualization:** Sarah G. Franklin, Maung Aung, Pauline E. Jolly.

**Data curation:** Omolade Adeniyi, LaTimberly Washington, Christina J. Glenn, Anniecia Scott, Soumya J. Niranjan.

**Formal analysis:** Christina J. Glenn, Soumya J. Niranjan.

**Funding acquisition:** Pauline E. Jolly.

**Investigation:** Omolade Adeniyi, LaTimberly Washington, Anniecia Scott, Maung Aung, Pauline E. Jolly.

**Methodology:** Omolade Adeniyi, LaTimberly Washington, Christina J. Glenn, Anniecia Scott, Maung Aung, Soumya J. Niranjan, Pauline E. Jolly.

**Project administration:** Maung Aung, Pauline E. Jolly.

**Resources:** Maung Aung.

**Software:** Christina J. Glenn.

**Supervision:** Maung Aung, Soumya J. Niranjan, Pauline E. Jolly.

**Visualization:** Sarah G. Franklin, Pauline E. Jolly.

**Writing – original draft:** Omolade Adeniyi, LaTimberly Washington, Sarah G. Franklin, Maung Aung, Soumya J. Niranjan, Pauline E. Jolly.

**Writing – review & editing:** Omolade Adeniyi, LaTimberly Washington, Christina J. Glenn, Sarah G. Franklin, Anniecia Scott, Soumya J. Niranjan, Pauline E. Jolly.

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
