## [Decision Letter · Decision Letter 0]

26 Aug 2020

PONE-D-20-21696

The Use of Complementary and Alternative Medicine among Hypertensive and Type 2 Diabetic Patients in Western Jamaica: A Mixed Methods Study

PLOS ONE

Dear Dr. Jolly,

Thank you for submitting your manuscript to PLOS ONE. After careful consideration, we feel that it has merit but does not fully meet PLOS ONE’s publication criteria as it currently stands. Therefore, we invite you to submit a revised version of the manuscript that addresses the points raised during the review process.

We look forward to receiving your revised manuscript.

Kind regards,

Jenny Wilkinson, PhD

Academic Editor

PLOS ONE

Journal Requirements:

2. Please address the following:

- Please include additional information regarding the interview guide and questionnaire used in the study and ensure that you have provided sufficient details that others could replicate the analyses. For instance, if you developed a questionnaire as part of this study and it is not under a copyright more restrictive than CC-BY, please include a copy, in both the original language and English, as Supporting Information. With regards to the questionnaire, please also provide further details concerning the development and validation of this tool.

- Please ensure you have discussed how the sample size was determined for the quantitative arm of this study, for example following a sample size calculation.

3.We note that you have indicated that data from this study are available upon request. PLOS only allows data to be available upon request if there are legal or ethical restrictions on sharing data publicly. For information on unacceptable data access restrictions, please see http://journals.plos.org/plosone/s/data-availability#loc-unacceptable-data-access-restrictions.

Additional Editor Comments (if provided):

Thank you for your submission, reviewers have provided comments and I now invite you to respond to these comments. I draw your attention particularly to the comments in relation to the methods and results as addressing these items will strengthen your work. Appropriate inferential statistic should be included in the data analysis.

In addition, please ensure that scientific names are given for all herbs that are mentioned and that capitalisation of drug and plant names are appropriate (i.e. capitals for trade names and lower case for common names of plants/herbs).

Reviewers' comments:

Reviewer's Responses to Questions

**Comments to the Author**

1. Is the manuscript technically sound, and do the data support the conclusions?

Reviewer #1: Yes

Reviewer #2: No

2. Has the statistical analysis been performed appropriately and rigorously? 

Reviewer #1: I Don't Know

Reviewer #2: I Don't Know

3. Have the authors made all data underlying the findings in their manuscript fully available?

Reviewer #1: No

Reviewer #2: No

4. Is the manuscript presented in an intelligible fashion and written in standard English?

Reviewer #1: Yes

Reviewer #2: No

5. Review Comments to the Author

Reviewer #1: i think this research does a good job highlighting the prevalent use of nontraditional medicines in the management of chronic health conditions and the importance of more explicit exploration of usage by patients on the part of health care providers. where is the data reporting on the different types of CAM being used by the surveyed patients? of course, the focus was obviously on the use of of herbal supplements/foods as representative of CAM (there is passing mention of exercise, spiritual healing, relaxation techniques, diet modification). the paragraph that begins at line 76 seems unnecessary.

Reviewer #2: i. The manuscript somehow sound, though some information not presented, e.g. sample size, sampling techniques and procedures, and the data may support the conclusion after addressing the comments. Also, to much limitations while some of them are in the Author’s control.

ii. The author didn’t indicate the statistical analysis program used to analyze the quantintaive data.

iii. The author is not ready to make the data available freely, some restrictions will apply

iv. The manuscript presented in an intelligible fashion, but some sentences are not clearly explained, so need some improvements in language.

v. Research ethics are already considered

vi. Other comments found in the manuscript attached to this document.

6. PLOS authors have the option to publish the peer review history of their article (what does this mean?). If published, this will include your full peer review and any attached files.

Reviewer #1: No

Reviewer #2: No

---

## [Author Response · Author response to Decision Letter 0]

3 Nov 2020

November 2, 2020

RE: PONE-D-20-21696

The Use of Complementary and Alternative Medicine among Hypertensive and Type 2 Diabetic Patients in Western Jamaica: A Mixed Methods Study

Dear Editor,

Thank you for sending the comments from the reviewers of our paper submitted to PLOS ONE. We have made the corrections requested and have attached the revised paper with highlights and a clean version for further consideration. We thank the reviewers for their careful review and believe that the changes have significantly improved the quality of the manuscript. This is a point-by-point response detailing the revisions that have been made and highlighted in the manuscript.

Journal Requirements:

Response: We have reviewed the style templates to ensure that the manuscript meets PLOS ONE's style requirements, including those for file naming.

2. Please include additional information regarding the interview guide and questionnaire used in the study and ensure that you have provided sufficient details that others could replicate the analyses. For instance, if you developed a questionnaire as part of this study and it is not under a copyright more restrictive than CC-BY, please include a copy, in both the original language and English, as Supporting Information. With regards to the questionnaire, please also provide further details concerning the development and validation of this tool. Please ensure you have discussed how the sample size was determined for the quantitative arm of this study, for example following a sample size calculation.

Response: We have added detailed information regarding development of the focus group interview guide and the questionnaire used in the study on pages 6-7 and have included a copy of each, in English (the original language) as Supporting Information. The quantitative results serve as evidence for hypothesis generation for future studies. Therefore, our participant sample size (N=60) is sufficient. 

b) If there are no restrictions, please upload the minimal anonymized data set necessary to replicate your study findings as either Supporting Information files or to a stable, public repository and provide us with the relevant URLs, DOIs, or accession numbers. Please see http://www.bmj.com/content/340/bmj.c181.long for guidelines on how to de-identify and prepare clinical data for publication. For a list of acceptable repositories, please see http://journals.plos.org/plosone/s/data-availability#loc-recommended-repositories. We will update your Data Availability statement on your behalf to reflect the information you provide.

Response: a) b) There are no ethical or legal restrictions on sharing a de-identified data set. The anonymized data set has been uploaded as a Supporting file titled “S1_Dataset.xlsx” with our revised submission

Response: We have included captions for the Supporting Information files at the end of the manuscript and have updated in-text citations to match accordingly.

Additional Editor Comments:

Thank you for your submission, reviewers have provided comments and I now invite you to respond to these comments. I draw your attention particularly to the comments in relation to the methods and results as addressing these items will strengthen your work. Appropriate inferential statistic should be included in the data analysis.

Response: We have addressed the comments related to the methods and results that strengthen the paper. Appropriate inferential statistics have been included in the data analysis and the new results presented in Tables 1-4. The text of the quantitative results has been revised accordingly.

In addition, please ensure that scientific names are given for all herbs that are mentioned and that capitalization of drug and plant names are appropriate (i.e. capitals for trade names and lower case for common names of plants/herbs). 

Response: This has been done.

5. Review Comments to the Author

Reviewer #1: 

i. I think this research does a good job highlighting the prevalent use of nontraditional medicines in the management of chronic health conditions and the importance of more explicit exploration of usage by patients on the part of health care providers. Where is the data reporting on the different types of CAM being used by the surveyed patients? of course, the focus was obviously on the use of herbal supplements/foods as representative of CAM (there is passing mention of exercise, spiritual healing, relaxation techniques, diet modification).

Response: We agree with the reviewer that the focus of the study was the use of herbal treatments. We do provide information on the sequential order of the different types of CAM used by the patients, i.e. herbal medicine followed by exercise, spiritual healing, relaxation techniques and diet modification on page 10 of the manuscript. We did not conduct further investigation into these different types of CAM in this study. 

ii. The paragraph that begins at line 76 seems unnecessary.

Response: We agree with the reviewer and have deleted the paragraph.

Reviewer #2: 

i. The manuscript somehow sound, though some information not presented, e.g. sample size, sampling techniques and procedures, and the data may support the conclusion after addressing the comments. 

Response: We have added details on the convenience sampling method and procedures used in the study on pages 5-6 of the manuscript. We have explained that the quantitative study was designed to collect data to serve as evidence for hypothesis generation for future studies. Therefore, a sample size of 60 participants was considered sufficient. 

ii. Also, too much limitations while some of them are in the Author’s control.

Response: We thank the reviewer and have revised the limitations. 

iii. The author didn’t indicate the statistical analysis program used to analyze the quantitative data.

Response: The statistical analysis method has been added to the paper as stated below. “The questionnaire data for the 60 participants were entered into excel and imported into JMP Pro 14.0 for analysis. Descriptive statistics were used to describe demographic characteristics of participants using mean ± standard deviation for continuous variables and frequency (percentage) for categorical variables. Demographic characteristics between disease groups were compared using a Fisher’s Exact test for categorical characteristics, and an analysis of variance for continuous characteristics. The significance level for these comparisons was set at p ≤0.05.”

iv. The author is not ready to make the data available freely, some restrictions will apply

Response: We have uploaded the dataset as a supporting file labeled “S1_Dataset.xlsx”.

v. The manuscript presented in an intelligible fashion, but some sentences are not clearly explained, so need some improvements in language.

Response: The manuscript has been revised thoroughly to clarify statements and improve the language.

vi. Research ethics are already considered

Response: Thank you.

vii. Other comments found in the manuscript attached to this document.

Response: We have addressed all of the comments in the manuscript.

Thank you for your kind consideration.

Respectfully, 

Pauline Jolly, PhD, MPH

Professor,

Director, UAB Minority Health International Research Training Program

Recipient, 2014 Ellen Gregg Ingalls/UAB National Alumni Society Award for Lifetime Achievement in Teaching 

2018 Fulbright Specialist Scholar, Institute of Public Health, Ho Chi Minh City, Vietnam

---

## [Decision Letter · Decision Letter 1]

1 Dec 2020

PONE-D-20-21696R1

The Use of Complementary and Alternative Medicine among Hypertensive and Type 2 Diabetic Patients in Western Jamaica: A Mixed Methods Study

PLOS ONE

Dear Dr. Jolly,

Thank you for submitting your manuscript to PLOS ONE. After careful consideration, we feel that it has merit but does not fully meet PLOS ONE’s publication criteria as it currently stands. Therefore, we invite you to submit a revised version of the manuscript that addresses the points raised during the review process.

We look forward to receiving your revised manuscript.

Kind regards,

Jenny Wilkinson, PhD

Academic Editor

PLOS ONE

Additional Editor Comments (if provided):

Thank you for your revisions. These have now been reviewed by the original reviewers and some further comments provided. I now invite you to provide a further response to these comments.

Reviewers' comments:

Reviewer's Responses to Questions

**Comments to the Author**

1. If the authors have adequately addressed your comments raised in a previous round of review and you feel that this manuscript is now acceptable for publication, you may indicate that here to bypass the “Comments to the Author” section, enter your conflict of interest statement in the “Confidential to Editor” section, and submit your "Accept" recommendation.

Reviewer #1: (No Response)

Reviewer #2: (No Response)

2. Is the manuscript technically sound, and do the data support the conclusions?

Reviewer #1: Yes

Reviewer #2: No

3. Has the statistical analysis been performed appropriately and rigorously? 

Reviewer #1: I Don't Know

Reviewer #2: Yes

4. Have the authors made all data underlying the findings in their manuscript fully available?

Reviewer #1: Yes

Reviewer #2: No

5. Is the manuscript presented in an intelligible fashion and written in standard English?

Reviewer #1: Yes

Reviewer #2: No

6. Review Comments to the Author

Reviewer #1: the sentence beginning line 66 can be rewritten so it is not run-on: "A survey conducted in 2000 examining use of herbal remedies from both urban and rural Jamaicans found that 100% of participants used herbs".

"advice" should be "advise" line 266.

"Warfarin" should be "warfarin" line 349.

i think the statements made in lines 367-368 and 395-397 are incorrect, participants did report experiences with and/or seem to suspect the prospect of synergistic effects of medications and herbs leading to hypotension and hypoglycemia.

Reviewer #2: Review Comments to the Author

Please use the space provided to explain your answers to the questions above. You may also include additional comments for the author, including concerns about dual publication, research ethics, or publication ethics

i. The manuscript will sound after addressing the comments.

• Sample size of 60 questionnaires and 25 FGD not enough, maybe Author should explain how is attained to that sample size (state the formula employed),

• Sampling techniques and procedures didn't clearly explain

• Some information presented in the results section while do not find in the table and the author declared that they are not there, so I don’t know why presented in the section while are not there.

ii. This work needs more information than what has been presented in the analyzed data, possibly the tool used missed some questions to grasp those information. For instance the author presented some information in the result text while are not found in the table and she/he declared that data not found.

iii. State when and for how long did you collect the data

iv. The manuscript needs an English native speaker to make it reader friendly and understandable.

v. The manuscript lacks novel part.

vi. Find other comments on the manuscript.

7. PLOS authors have the option to publish the peer review history of their article (what does this mean?). If published, this will include your full peer review and any attached files.

Reviewer #1: No

Reviewer #2: No

---

## [Author Response · Author response to Decision Letter 1]

22 Dec 2020

December 21, 2020 

RE: PONE-D-20-21696

The Use of Complementary and Alternative Medicine among Hypertensive and Type 2 Diabetic Patients in Western Jamaica: A Mixed Methods Study

Jenny Wilkinson, PhD

Dear Dr. Wilkinson,

Thank you for sending the second set of comments from the reviewers of our paper submitted to PLOS ONE. We have made the corrections requested by the academic editor and the reviewer(s) and have attached a highlighted copy of the manuscript that highlights changes made to the original version labeled as 'Revised Manuscript with Track Changes' and an unmarked version of the revised paper labeled 'Manuscript'. This is a point-by-point Response to Reviewers detailing the revisions that have been made and highlighted in the manuscript.

Reviewers' comments:

Reviewer's Responses to Questions

Comments to the Author

1. If the authors have adequately addressed your comments raised in a previous round of review and you feel that this manuscript is now acceptable for publication, you may indicate that here to bypass the “Comments to the Author” section, enter your conflict of interest statement in the “Confidential to Editor” section, and submit your "Accept" recommendation.

Reviewer #1: (No Response)

Reviewer #2: (No Response)

 2. Is the manuscript technically sound, and do the data support the conclusions?

 Reviewer #1: Yes

Reviewer #2: No

 3. Has the statistical analysis been performed appropriately and rigorously? 

 Reviewer #1: I Don't Know

Reviewer #2: Yes

4. Have the authors made all data underlying the findings in their manuscript fully available?

 Reviewer #1: Yes

Reviewer #2: No

 5. Is the manuscript presented in an intelligible fashion and written in standard English?

Reviewer #1: Yes

Reviewer #2: No

 6. Review Comments to the Author

Reviewer #1: the sentence beginning line 66 can be rewritten so it is not run-on: "A survey conducted in 2000 examining use of herbal remedies from both urban and rural Jamaicans found that 100% of participants used herbs". 

Response: We thank the reviewer and have revised the sentence to read “A survey that examined the use of herbal remedies among rural and urban Jamaicans of varying socioeconomic groups found that 100% of the participants used herbs [8].” 

"advice" should be "advise" line 266.

Response: We thank the reviewer and have made this correction. 

"Warfarin" should be "warfarin" line 349.

Response: We have made this correction. 

i think the statements made in lines 367-368 and 395-397 are incorrect, participants did report experiences with and/or seem to suspect the prospect of synergistic effects of medications and herbs leading to hypotension and hypoglycemia.

Response: We thank the reviewer for this observation and have deleted these statements.

Reviewer #2: Review Comments to the Author

Please use the space provided to explain your answers to the questions above. You may also include additional comments for the author, including concerns about dual publication, research ethics, or publication ethics

i. The manuscript will sound after addressing the comments.

• Sample size of 60 questionnaires and 25 FGD not enough, maybe Author should explain how is attained to that sample size (state the formula employed),

• Sampling techniques and procedures didn't clearly explain

Response: We have revised the explanation in the methods section (page 5) and added references to justify the sample sizes as follows: “An exploratory design was used for the quantitative portion of this study (Creswell, J. W. and V. L. Plano Clark (2018). Prior studies suggest the sample size (N=60) is sufficient since the exploratory nature of the quantitative survey is the first stage of data collection and provides a rationale for defining future hypotheses for other stages of study (Kutner et al. 1999). In the concurrent quantitative-qualitative design, a smaller qualitative data design sample (N=25) was determined as sufficient (Creswell, J. W. and V. L. Plano Clark (2018). 

References:

• Kutner JS, Steiner JF, Corbett KK, Jahnigen DW, Barton PL. Information needs in terminal illness. Soc Sci Med. 1999;48(10):1341-1352.

• Creswell JW, Clark VLP. Designing and conducting mixed methods research. 3rd edition. SAGE Publications, Inc. 2018

• Some information presented in the results section while do not find in the table and the author declared that they are not there, so I don’t know why presented in the section while are not there.

ii. This work needs more information than what has been presented in the analyzed data, possibly the tool used missed some questions to grasp those information. For instance the author presented some information in the result text while are not found in the table and she/he declared that data not found. 

Response: We have added a table (Table 3) to the paper that shows the total number of participants who selected each herb and total numbers stratified by disease group. These data are for reporting only and not for analysis, so no data analysis was conducted and no data were missed (pages 11-12). 

iii. State when and for how long did you collect the data

Response: We have added the dates of the study as May to August 2018 to both the abstract and methods (page 4).

iv. The manuscript needs an English native speaker to make it reader friendly and understandable.

Response: We are native English speakers and professionals. We have re-read and revised the paper to make it more understandable.

v. The manuscript lacks novel part.

Response: This is the first mixed-methods study on CAM use by HTN and T2DM patients in western Jamaica and provides the basis for future studies and interventions on alternative treatments. We agree with the reviewer that we did not expressly highlight the novelty of this study. We have revised the conclusion of the abstract to read “This study is novel in that it provides useful insights into perceptions and use of alternative treatments by patients that can be used by HCPs in developing appropriate interventions to encourage proper use of prescription medicines and CAM resulting in improved management of these chronic diseases” (pages 2-3).

We have also revised the Conclusion section of the paper to highlight that “the findings from this study indicate the need to include salient information on CAM in the professional curricula of HCPs and can be used to develop appropriate interventions to ensure the proper use of prescription medicines and CAM. This should result in improved management of T2DM and HTN among patients” (page 23).

vi. Find other comments on the manuscript.

Response:

 7. PLOS authors have the option to publish the peer review history of their article (what does this mean?). If published, this will include your full peer review and any attached files.

Do you want your identity to be public for this peer review? For information about this choice, including consent withdrawal, please see our Privacy Policy.

 Reviewer #1: No

Reviewer #2: No

 Thank you for your kind consideration.

Respectfully, 

Pauline Jolly, PhD, MPH

Professor,

Director, UAB Minority Health International Research Training Program

Recipient, 2014 Ellen Gregg Ingalls/UAB National Alumni Society Award for Lifetime Achievement in Teaching 

2018 Fulbright Specialist Scholar, Institute of Public Health, Ho Chi Minh City, Vietnam

---

## [Editor Report · Decision Letter 2]

23 Dec 2020

The Use of Complementary and Alternative Medicine among Hypertensive and Type 2 Diabetic Patients in Western Jamaica: A Mixed Methods Study

PONE-D-20-21696R2

Dear Dr. Jolly,

We’re pleased to inform you that your manuscript has been judged scientifically suitable for publication and will be formally accepted for publication once it meets all outstanding technical requirements.

Kind regards,

Jenny Wilkinson, PhD

Academic Editor

PLOS ONE

Additional Editor Comments (optional):

Thank you for responding to reviewer comments and revising your manuscript, these have satisfactorily addressed the issues raised.
---

## [Editor Report · Acceptance letter]

6 Jan 2021

PONE-D-20-21696R2 

The Use of Complementary and Alternative Medicine among Hypertensive and Type 2 Diabetic Patients in Western Jamaica: A Mixed Methods Study 

Dear Dr. Jolly:

I'm pleased to inform you that your manuscript has been deemed suitable for publication in PLOS ONE. Congratulations! Your manuscript is now with our production department. 

Kind regards, 

on behalf of

Dr Jenny Wilkinson 

Academic Editor

PLOS ONE